# Nuclear Mechanotransduction in Skeletal Muscle

**DOI:** 10.3390/cells10020318

**Published:** 2021-02-04

**Authors:** Saline Jabre, Walid Hleihel, Catherine Coirault

**Affiliations:** 1Sorbonne Université, INSERM UMRS-974 and Institut de Myologie, 75013 Paris, France; saline.j.jabr@net.usek.edu.lb; 2Department of Biology, Faculty of Arts and Sciences, Holy Spirit University of Kasik (USEK), Jounieh 446, Lebanon; walidhleihel@usek.edu.lb; 3Department of Basic Health Sciences, Faculty of Medicine, Holy Spirit University of Kaslik (USEK), Jounieh 446, Lebanon

**Keywords:** mechanotransduction, muscle disorders, nucleus, nucleo-cytoplasmic coupling, mechanics

## Abstract

Skeletal muscle is composed of multinucleated, mature muscle cells (myofibers) responsible for contraction, and a resident pool of mononucleated muscle cell precursors (MCPs), that are maintained in a quiescent state in homeostatic conditions. Skeletal muscle is remarkable in its ability to adapt to mechanical constraints, a property referred as muscle plasticity and mediated by both MCPs and myofibers. An emerging body of literature supports the notion that muscle plasticity is critically dependent upon nuclear mechanotransduction, which is transduction of exterior physical forces into the nucleus to generate a biological response. Mechanical loading induces nuclear deformation, changes in the nuclear lamina organization, chromatin condensation state, and cell signaling, which ultimately impacts myogenic cell fate decisions. This review summarizes contemporary insights into the mechanisms underlying nuclear force transmission in MCPs and myofibers. We discuss how the cytoskeleton and nuclear reorganizations during myogenic differentiation may affect force transmission and nuclear mechanotransduction. We also discuss how to apply these findings in the context of muscular disorders. Finally, we highlight current gaps in knowledge and opportunities for further research in the field.

## 1. Introduction

Skeletal muscle is a highly organized tissue designed to produce force and movement. It is composed of differentiated, multinucleated and aligned myofibers responsible for contraction, and also contains a population of mononucleated muscle cell precursors (MCPs), that are maintained in a quiescent state under homeostatic conditions. Fusion of tens of thousands of differentiated MCPs (myocytes) produces multinucleated myotubes which mature into myofibers, composed of a regular array of contractile elements, the sarcomere [1]. Skeletal muscle is remarkable in its ability to adapt in response to the demands imposed on it, a property referred to as muscle plasticity. Low physical activity and some disease conditions lead to the reduction in myofiber size, called atrophy, whereas hypertrophy refers to the increase in myofiber size induced by high physical activity or intrinsic factors such as anabolic hormones/drugs. Molecular mechanisms that regulate changes in skeletal muscle mass in response to mechanical load have been detailed [2,3,4,5]. In post-mitotic muscle cells, mechanical loading impacts translational events, thereby regulating the rate of protein synthesis leading to changes in myofibrillar protein content [2]. In addition, mechanical loading triggers changes in the cell cycle rate [6,7] and MCP proliferation [5]. The fusion of MCPs to the growing fiber allows the addition of new myonuclei, which are likely to contribute to sustained and harmonious muscle growth [8]. Finally, the nucleus triggers diverse cell responses in response to nuclear envelope deformation: nuclear accumulation of the transcription factors yes-associated protein (YAP)/transcriptional co-activator with PDZ-binding motif (TAZ) [6,9,10], activation of the ataxia telangiectasia and Rad3-related protein kinase [11,12], calcium release [13], activation of the calcium-dependent cytosolic phospholipase A2 [14] and rupture of the nuclear envelope (NE) associated with DNA damage [15,16].

The nucleus is generally the stiffest element of all eukaryotic cells [17]. In addition to being the site for storage of genetic material and gene transcription, the nucleus plays crucial roles in mechanotransduction, which is the transduction of exterior physical forces to generate a biological response [18]. Nuclear mechanotransduction is likely to play important roles in skeletal muscle physiology and adaptation. Force transmission from the cell periphery to the nucleus involves the cytoskeleton, the LINC complex (Linker of Nucleoskeleton and Cytoskeleton) and the proteins associated with the NE, including emerin and lamins. Mechanical force induces changes in nuclear lamina polymerization and chromatin condensation state, thereby regulating translational capacity and efficiency, nuclear elasticity, and deformability [2,19,20,21,22,23] and in turn, the cell response to mechanical stress.

Nuclear mechanotransduction is essential to help the muscle to adapt in response to changes in physical activity [4,24] or in mechanical stimuli arising from the surrounding extracellular matrix or from neighboring cells [25]. Numerous studies have gained insights into the molecular mechanisms associated with muscle mechanotransduction and their role in skeletal muscle growth [26,27,28,29,30,31]. The role of the cytoskeleton in regulating nuclear shape via interaction with the NE has been detailed in different cell types including muscle cells [32,33,34]. Interestingly, cytoskeleton and nuclear architectures are dynamically regulated. They respond to the mechanical environment and differ according to the myogenic state [34]. In addition, signaling molecules and transcription factors such as YAP, TAZ, and serum responsive factor have emerged as important signaling pathways to relay mechanical signals and regulate dynamics of cytoskeleton, gene expression, and in turn myogenic development of striated muscle [28,31,35,36,37,38]. Importantly, because intracellular structures and signaling pathways are developmentally regulated, the myogenic process is likely to modulate in turn the nuclear mechanotransduction, thus differentially modulating the force response on MCPs, myotubes and terminally differentiated myofibers. Finally, direct or indirect mechanisms responsible for defective cytoskeleton and nuclear architectures are likely to impact the nuclear response and contribute to muscle dysfunction in muscle diseases.

In this review, the cellular and molecular mechanisms regulating nuclear mechanotransduction in skeletal muscle are updated, and findings regarding nuclear force transmission and nuclear response to mechanical forces in MCPs and multinucleated myofibers are summarized. Based on data from diverse cell types including myogenic cells, we will focus on how myogenic differentiation can affect force transmission to the nucleus. Finally, we will discuss how to apply these findings in the context of muscular disorders.

## 2. Cytoskeletal Components Relevant for Force Transmission to the Nucleus

The cytoskeletal components relevant for force transmission to the nucleus include actin filaments (F-actin), microtubules (MTs) and intermediate filaments (IFs), whose structural and functional organization, including assembly sites, dynamics, turnover and integration with other cell components, determine function [39,40,41]. The perinuclear cytoskeleton provides a structural network to transmit and focus pushing or pulling forces onto the nucleus [40,42] through specialized proteins that comprise the LINC complex [43,44,45]. The amount and organization of the cytoskeletal and LINC components are tissue-specific and developmentally regulated (see below).

Major reorganization of the cytoskeleton network occurs during the process of muscle differentiation (Figure 1), with functional consequences on force transmission to the nuclear envelope and thus on the nuclear response. Although force transmission to the nucleus is crucial for MCP fate, a major contribution of the distribution of the cytoskeleton in mature striated muscle fibers could be to transmit force to the extracellular matrix (ECM) while protecting myonuclei from the axial contractile force generated by the contractile apparatus.

### 2.1. The Perinuclear Actin Network and Muscle Differentiation

In different cell types, perinuclear actin emerges as a critical component for proper nucleo-cytoskeletal connections [39,40,46]. On the dorsal side of the nucleus of cells grown in 2D culture, perinuclear actin comprises the actin cap formed by dorsal stress fibers [47] (Figure 2A) and the so-called transmembrane actin-associated nuclear (TAN) lines [48] (Figure 2B,C). The actin cap is composed of thick parallel and highly contractile acto-myosin filaments, tightly connected to the nucleus, and attached to basal focal adhesion sites on both extremities [47,49,50,51]. The perinuclear actin cap accumulates upon mechanical stimulation [49,50] and has important roles in nuclear mechanotransduction [50,52].

The actin cap is developmentally regulated, being present in myoblasts but absent in undifferentiated embryonic stem cells [53] and terminally differentiated muscle cells [27]. The structural and functional organization of actin cytoskeleton in the perinuclear region of myotubes remain partly unknown. During skeletal myofiber formation, nuclei are initially in the center of the myofiber and then move towards to myofiber periphery [54]. It has been shown that amphiphysin-2/BIN1, which is mutated in centronuclear myopathies, triggers peripheral nuclear positioning to the periphery of myofibers via N-WASP and actin, thus implicating the actin cytoskeleton in nuclear movement [55]. In addition, perinuclear actin may significantly alter the nuclear shape [27]. However, nuclear positioning to the myofiber periphery is mediated by centripetal forces arising from myofibril contraction around the nucleus [27]. Furthermore, it has been proposed that a nucleus–cytoskeleton connection is not required for peripheral nuclear movement [27]. Future work should address how structural and functional connections between perinuclear actin network and nuclei are modified during skeletal myofiber formation. In addition to extensive cytoskeletal reorganization, shifts in expression of actin components from non-muscle to muscle isoforms occur during skeletal myogenesis [56,57,58]. The muscle-specific isoform α-actin becomes the predominant actin in terminally differentiated myofibers and localizes to the sarcomeric thin filaments, where it interacts with myosin to produce a contractile force [59,60]. The non-muscle actins γ and β that are present around the nucleus in myoblasts [61] are downregulated during terminal differentiation of myoblasts into myotubes. In terminally differentiated myofibers, γ- and β-actins reside in the cortical cytoskeleton and at costameres [62,63,64,65]. The costameric F-actin network is thought to contribute with other proteins to the radial transmission of contractile force outward from the sarcomere to the extracellular matrix, adjacent muscle fibers, and beyond [64]. Therefore, non-muscle F-actin could serve opposite force transmission direction according to the state of myogenic differentiation. The direction could be predominantly external to internal, toward NE in myoblasts, but predominantly internal and sarcomeric to external, toward extracellular matrix, in myofibers (Figure 1).

### 2.2. The MTs

MTs are three orders of magnitude stiffer than actin, IFs being the softness among the three major types of cytoskeleton filaments [65]. Their radial, centrosome-dominated distribution in myoblasts [66,67] may favor the transmission of external mechanical forces to the NE and influence nuclear shape [68] and function [69] (Figure 1A). During the differentiation process, there is a large reorganization of the centrosome proteins: myoblasts possess a morphologically recognizable centrosome with characteristic marker proteins concentrated in the pericentriolar material, whereas myotube differentiation requires relocalization of centrosome proteins to the surface of the nucleus [67,70,71]. Centrosome proteins are critical for MT nucleation and/or anchoring; therefore, MT orientation is extensively redistributed into a more ordered paraxial array in myotubes [66,67,72,73] (Figure 1B). Mature myofibers also exhibit a perinuclear network of MTs, comprising a cage-like structure of a high-density meshwork that may be responsible for nuclear shaping and mechanical protection, and a circular and radial-anisotropic MTs, which are either polarized in the direction of contraction or in the lateral direction [74]. MT post-translational modifications such as increased detyrosinated [75,76] and binding of MTs to MT-associated proteins (MAPs), including EB1 and spectraplakin [74], confers stability to the MTs and has been shown to be essential for maintaining myonuclear morphology [74]. Additionally, it has been proposed that the spectrin domains of nesprin confers elastic features of the MT–spectraplakin–EB1 perinuclear network during the contraction of striated muscle [74]. As a consequence, primary defects in the nuclear-associated networks of MTs have been implicated in strain-induced myonuclear damage [27,74,77].

### 2.3. Cytoplasmic IFs

IFs have emerged as a perfect candidate for maintaining proper nuclear mechano-response because they are able to resist high mechanical stresses, i.e., bending and stretching, to a considerable degree [65]. IFs are surprisingly flexible [78,79,80,81,82] and can undergo strain-stiffening [83,84,85]. This is due to the short persistence length of intermediate filaments (1–3 μm) [65]. In the cytoplasm, they can form mechanically relevant links to each other, to other cytoskeletal filaments, to membrane complexes, and to internal organelles including the nucleus [82,86] (Figure 1). These mechanical properties and interconnections enable the IFs to serve as mechanical stress absorbers that protect the cytoplasm and organelles, including the nucleus, against large deformations [51,87,88]. This idea is supported by the fact that IFs can withstand deformations of up to 300% of their initial length without rupturing [89]. Several IFs are expressed and developmentally regulated in human skeletal muscle cells [90,91,92,93]. Non muscle-specific proteins vimentin and nestin are expressed in MCPs and myoblasts and are downregulated during later differentiation [94]. Desmin, the muscle-specific IF protein, is expressed at low levels in MCPs and its expression continuously increases to become the prominent IF in mature myofibers [94,95]. It can form copolymers with synemin, another non-muscle specific IF, around the α-actinin-rich Z-lines [92]. In undifferentiated myoblasts, vimentin and desmin are stably linked to the outer nuclear membrane [96] via plectin [97], thus contributing to the perinuclear cage-like structure. During terminal muscle differentiation, desmin accumulates and forms a three-dimensional network between the contractile apparatus, the extracellular matrix, and other cell organelles such as mitochondria, T-tubules, and nuclei [95,98,99,100] (Figure 1). Close to the nucleus, desmin filaments extend from the Z-lines of striated muscles towards the NE, where they interact with plectin. Terminal differentiation-induced desmin redistribution is associated with post-translational modifications such as phosphorylation and ADP-ribosylation [101], which in turn regulate IF assembly and disassembly as well as interactions between IFs and other cell components and structures [102]. In mature muscle fibers, the primary role of desmin is to link adjacent myofibrils to each other and to the extracellular matrix, via costameres [39,103,104,105]. Consequently, a functional reduction in desmin is associated with structural instability of the sarcomeres [106]. Accumulating evidence indicates that desmin is also crucial as a stress-transmitting and stress-signaling network [98,107,108,109,110]. Desmin interactions with the nucleus are required to maintain nuclear architecture in cardiomyocytes [111] and to prevent nuclear and muscle damage in response to mechanical challenges [111,112]. Future studies will determine the contribution of desmin scaffolds in myonucleus architecture and function.

## 3. Mechanical Linkages between the Cytoskeleton and the Nucleoskeleton

LINC complexes provide direct physical nucleo-cytoskeletal coupling between the cytoskeleton network and the NE [113,114] (Figure 3). The LINC complexes comprise outer nuclear transmembrane proteins, called nesprins (NE Spectrin-Repeat Proteins) defined by the Klarsicht-ANC1-Syne-homology (KASH) domain. This domain directly interacts with the luminal domain of the inner nuclear membrane proteins Sad1 and UNC-84 (SUN) proteins 1 (SUN1) or 2 (SUN2) [44,113] within the perinuclear space of the nuclear envelope. SUN proteins form trimers and span the inner nuclear membrane, with their N-amino-terminal nucleoplasmic domains interacting with lamins and lamin-associated proteins within the nucleoplasm [115]. By crossing the outer nuclear membrane, nesprins provide a mechanical link from the cytoskeleton to the nucleoskeleton.

To date, six genes encoding for different nesprins (-1,-2,-3,-4, lymphoid-restricted membrane protein (LRMP) and KASH5) have been identified in mammals [97,116,117]. Giant nesprins-1 and -2 are ubiquitously expressed with highest representation in striated muscle [118,119]. The *SYNE-1* and *SYNE-2* genes encode the nesprin-1G (1008 kD) and nesprin-2G (792 kD), respectively, with calponin domains at their N-termini that bind the actin cytoskeleton [116]. Nesprins-1G and -2G also bind to the MT motors dynein and kinesin via their C-terminal cytoplasmic stretch [113,120,121,122]. Kinesin-1 interacts with nesprin-1G and -2 via their LEWD motifs [119,120].

*SYNE-1* and *SYNE-2* have multiple internal promotors giving rise to shorter nesprin isoforms which lack the actin-binding domain [119,123] (Figure 3). Alternative splicing also generates short isoforms that lack the C-terminal KASH domain as well as short isoforms that lack both the KASH domain and CH domains [124].

In contrast to SUN proteins, nesprins-1 and -2 switch localizations and isoforms during myogenesis [118,119]. Nesprin-1 increases at the nuclear rim during early myogenesis but is partially replaced by nesprin-2 at later stages of muscle development [118,119]. However, nesprin-1 appears to be critical in synaptic and non-synaptic myonuclear anchoring in skeletal muscle [125,126], due to its ability to form interactions between myonuclei and actin cytoskeleton [125,126,127]. Expression of two shorter α isoforms, nesprin-1α2 and nesprin-2α1, is switched on during myogenesis [121,122,128] and becomes dominant in mature skeletal muscle [118]. They are found almost exclusively in skeletal and cardiac muscle [122,128] and form a complex with emerin and A-type lamins at the inner nuclear membrane [129,130]. At the outer nuclear membrane, nesprin-1α2 and nesprin-2α1 can interact with kinesin and microtubules [119,123] (Figure 3). Nesprin1-α2 is the main short form of nesprin-1 in skeletal muscle [131]. It is located mainly at the nuclear rim in early myotubes and immature muscle fibers, but then declines in most mature, adult muscle fibers [131], being restricted to neuromuscular junction nuclei [116,119]. Nesprin1-α2 is required for the correct positioning of myonuclei [77,120,132,133] and MT nucleation from the NE [119], by recruiting A-Kinase Anchoring Protein-450 to the NE [77]. Nesprin-3 lacks actin-binding domains but can indirectly connect to the cytoskeleton by binding to another protein with tandem actin-binding calponin homology domain [134]. Although nesprin-3 exists as two isoforms, nesprin-3α and nesprin-3β, only nesprin-3α can attach to the cytoskeleton. For instance, nesprin-3α can anchor IFs to the NE through plectin [121,122,123,126], a plakin family member that can also interact with actin filaments and MTs [97,135,136,137]. This plectin–nesprin interaction requires the dimerization of plectin and takes place between the N-terminal actin-binding domain of plectin and the first spectrin repeat of nesprin-3α [135]. Nesprin-3β does not interact with IFs because it lacks this spectrin-like repeat of nesprin-3α [135].

The different components of the LINC complexes have been associated with a number of pathogenic modifications in humans as well as in animal models. Perturbation of LINC complexes induces defective signal transduction across the NE [138,139], and prevents centrosome reorientation [48], chromatin organization [77,140,141,142,143], and abnormal nuclear positioning [116,121,131,144,145,146]. It has been shown that mutations in nesprins-1 and -2 cause Emery–Dreifuss muscular dystrophy [77,125,147,148,149,150] and dilated cardiomyopathy [149]. It has been proposed that the giant nesprin-1 regulates a feedback loop by which MCPs adapt their intracellular tension to the softness of their native extracellular microenvironment through nucleo-cytoskeletal connections [150]. In addition, nesprin mutations can impair the interaction of nesprin with lamins, emerin and/or SUN proteins, thus affecting diverse functions including gene expression, nuclear shape and positioning [149]. As yet, no mutation in nesprin-3 has been found to be responsible for skeletal muscle diseases. However, acute depletion of nesprin-3 does lead to rapid shrinkage and unfolding of nuclei in a microtubule-dependent manner in rat ventricular cardiomyocytes [111]. Loss of nuclear integrity is concomitant with compromised contractile function and has been proposed to contribute to the pathophysiological changes observed in desmin-related myopathies [111]. Further investigations are required to elucidate the complex mechanisms behind LINC-mediated nucleo-cytoskeletal linkages in skeletal muscle. Finally, although LINC complexes are critical for force transmission across the NE, alternative LINC-independent mechanisms have also been proposed [151]. For instance, it has been proposed that cell boundaries can drive nuclear flattening during cell spreading on rigid substrates [152]. It was shown that a direct compressive force by LINC-anchored apical actin cables is not required for nuclear flattening [152]. According to this model, the overall nuclear shape is primarily dictated by passive forces generated within the actin cytoskeleton, with cell spreading and forces transmitted by the actin cap or LINC complexes contribute to a lesser degree [151,153].

## 4. The Nuclear Lamina

The nuclear lamina is a filamentous network of proteins mainly composed of the type V IF lamin proteins that assemble into a meshwork underneath the inner nuclear membrane [154,155]. The lamina is composed of lamins and lamin-associated proteins and provides structural support to the NE [156]. Lamins can be categorized as A-type (lamin A/C) or B-type (lamin B1, B2) lamins. They are key components of the nuclear environment and interact with a large number of proteins [140,157,158,159], the nuclear membrane, and chromatin [157,160] to influence mechanical cues and signaling pathways crucial for cellular proliferation and differentiation [161]. In addition, lamins are involved in the epigenetic regulation of chromatin with drastic consequences for gene regulation [162].

The B-type lamins, lamins B1 and B2, coded for by the *LMNB1* and *LMNB2* genes, are expressed in all somatic cells. B-type lamins have an important role in nuclear shape [86,163] and structure [155,164,165] and may provide nuclear elastic resistance [164], particularly in cells with low A-type lamins [86,163,166]. However, B-type lamin expression differs minimally across solid tissues or in response to matrix stiffness [167] and does not appear to play a major role in nuclear stiffness [86], In contrast, A-type lamins, encoded for by the *LMNA* gene, are critical for the appropriate nucleus stiffening [166] and dictate the nuclear strain stiffening that dominates nuclear resistance to large deformations [20]. Indeed, upon nuclear mechanostimulation, nucleoplasmic domain of the inner nuclear membrane protein emerin becomes phosphorylated by the protein proto-oncogene tyrosine protein kinase Sarcoma (Src) [168,169]. The Ig fold domain of lamin A is able to partially unfold, leading to stretching of the protein [170]. A-type lamins undergoes dephosphorylation of the S22 residue, associated with relocalization of the nucleoplasmic fraction to the nuclear lamina [166,168,171]. This in turn reinforces the nuclear lamina by stabilization and assembly of A-type lamins and increases nuclear stiffness [161,166]. Conversely, in reduced mechanical constraints, the mobility and turnover of A-type lamins increases [166,171,172]. It has also been shown that, under compression, the coiled coils in the rod domains of A-type lamin polymers are able to slide over each other to contract the length of the rod, behaving as a compression spring able to absorb pressure [173]. The expression of A-type lamins can be correlated with tissue stiffness [166], stiff tissues such as muscle having higher A-type lamin expression and stiffer nuclei than those in softer tissues such as brain [166]. Moreover, the expression and stability of A-type lamins increase during myogenic differentiation [86], leading to nuclear stiffening [174].

Importantly, force-induced remodeling of the nuclear lamina may affect gene transcription by changing the binding properties of NE proteins and transcription factors. Indeed, it is known that chromatin containing actively transcribed genes exists in a less condensed state (i.e., euchromatin) compared to the more compact regions (i.e., heterochromatin) that contain silent genes. Chromatin contained in lamin-associated domains (LADs) is generally heterochromatin [175]. Force changes trigger rapid reorganization of the heterochromatin at the nuclear lamina and are associated with changes in global patterns of gene expression [176]. Nuclear stretch decreases the levels of repressive histone H3K9me3 at the nuclear periphery and increases chromatin mobility [177]. According to studies from Wickström’s lab, this chromatin response relies on ER Ca^2+^ release [22]. A-type lamin levels and nuclear stiffness determine the sensitivity of the ER calcium release, where stiffer nuclei are more prone to respond [22]. Interestingly, myogenic differentiation is associated with specific developmental gene repositioning to and from the nuclear periphery, generally associated with the repression of genes inhibitory to myogenesis and the activation of genes required for myotube differentiation [178]. Muscle-specific NE transmembrane proteins (NETs), including NET39, Transmembrane Protein 38A, and wolframin ER transmembrane glycoprotein, direct specific myogenic genes to the nuclear periphery to facilitate their repression and their combined knockdown almost completely blocks myotube formation [178]. There is also evidence that the disrupted tethering of myogenic genes with NE [169,170] and muscle-specific NETs [178] could underlie muscle pathology in NE-linked diseases. Alternatively, NET-directed gene repositioning may contribute to nuclear stiffening during differentiation.

In line with these physiological roles of A-type lamins, mutations in the *LMNA* gene cause laminopathies, a heterogeneous group of disorders, including skeletal muscle dystrophies and cardiomyopathies [156,179,180,181,182]. The severity of the muscle disease is highly variable, the most severe form being the *LMNA*-related congenital muscular dystrophy [183,184]. Although the physiopathology of the disease still requires further studies, there is clear evidence that impaired integrity of the nucleus [184,185,186,187,188], aberrant positioning of myogenic genes [178,189,190] and defective mechanotransduction signaling [29,31,185,191,192] all contribute to the muscle diseases related to *LMNA* mutations. Future studies will precisely determine how the combination of mechanical uncoupling/epigenic factors and a signaling defect could drive these skeletal muscle disorders.

## 5. Chromatin-Mediated Mechanoresponse

Whereas the lamina has been recognized for many years as a major contributor to nuclear stiffness, there is now evidence that chromatin and its histone modification state also contribute to nuclear mechanics independently of A-type lamins [20,23,193,194,195,196]. It has been proposed that chromatin dominates nuclear force responses at short extensions of <30% strain [20]. Chromatin-based nuclear rigidity operates by inducing changes in histone modification state. Alterations that produce more euchromatin or heterochromatin result in decreased or increased small extension nuclear stiffness, respectively [20].

Upon mechanical stimulation, untethering LADs from the nuclear lamina could initiated gene repositioning and transcription. Mechanical forces could also decondense gene loci at the nuclear periphery, thus allowing better access for transcription machinery and increased transcription. However, it is important to remember that genes located at the nuclear envelope are not necessarily silent [197,198,199], and that untethering from the lamina is not sufficient to induce changes in gene transcription [200,201]. Taking into account these limitations, there is evidence that force can induce chromatin rearrangement and gene activation. Indeed, the activation and transcription of many genes has been associated with effective force transmission to the nucleus and/or to nuclear deformations [184,202,203,204,205]. In addition, force-induced chromatin reorganization could play a critical role in stem cell differentiation [178,206,207]. Interestingly, data show that forces propagate through lamina–chromatin interactions to directly stretch the chromatin and induce transcription upregulation in a living cell [208]. How the altered chromatin-mediated mechanoresponse contributes to mechanical load-mediated adaptation in normal and pathological skeletal muscle remains open for future studies.

## 6. Nuclear Positioning and Mechanotransduction

Skeletal muscle fibers contain hundreds of flattened myonuclei evenly distributed at the periphery of each cell, with 3–8 nuclei (synaptic nuclei) anchored beneath the neuromuscular junction. How nuclei properly position themselves within each muscle fiber remains partly obscured, especially in tissues. Myonuclear positioning in skeletal muscle cells is an active process that occurs during the differentiation and maturation process, as well as during regeneration [209]. It involves the cytoskeletal network of MTs, F-actin and/or IFs as follows: MTs in the initial translocation/spacing of nuclei along the fiber [54,55,77,138,210], and F-actin and desmin in their movement to the fiber periphery [27]. Mislocalization of myonuclei has been associated with a variety of muscle disorders, characterized by muscle atrophy, muscle weakness, and reduced muscle performance [209,211].

The unique distribution of myonuclei at the muscle fiber periphery raises questions about the amount of intracellular force transmitted from the cytoskeleton to NE. Mispositioned myonuclei within individual multinucleated muscle fibers are a hallmark of many muscle diseases, including congenital myopathies and muscular dystrophies [55,125,138,210,212]. Abnormal nuclear positioning is likely to affect individual myonuclear activity by affecting force and strain transmission across the NE [74]. It has been proposed that centrally located myonuclei may experience higher contractile forces exerted by the myofibrils around the nucleus than peripheral nuclei which could disturb nuclear stability. However, whether or not mispositioned myonuclei are a cause or consequence of muscle disease states still remains to be determined.

## 7. Conclusions and Future Directions

An increasing number of studies focusing on the importance of appropriate nuclear mechanotransduction for muscle homeostasis, regeneration, and plasticity have appeared in the literature since 2010. Advances in deciphering the molecular mechanisms contributing to nuclear mechanotransduction strongly support the idea that defects in nuclear mechanotransduction contribute to human muscle disorders. However, an understanding of the mechanistic and physiological outcomes for nuclear mechanical stress response mainly arises from studies conducted in embryonic and/or mononucleated cells and may depend on the specific cell lines used. The majority of nuclear and cytoskeletal components involved in nuclear mechanotransduction are developmentally regulated and largely reorganized during muscle differentiation, which complicates the understanding of nuclear mechanotransduction defects in muscle disorders.

We anticipate that future research efforts will provide new insights into how the terminal differentiation of MCPs into multinucleated muscle fibers affects nuclear mechanotransduction. In addition, we foresee the elucidation of the contributive role of stress- and strain-induced nuclear response in normal and diseased striated muscles in the future.

## Figures and Tables

**Figure 1 cells-10-00318-f001:**
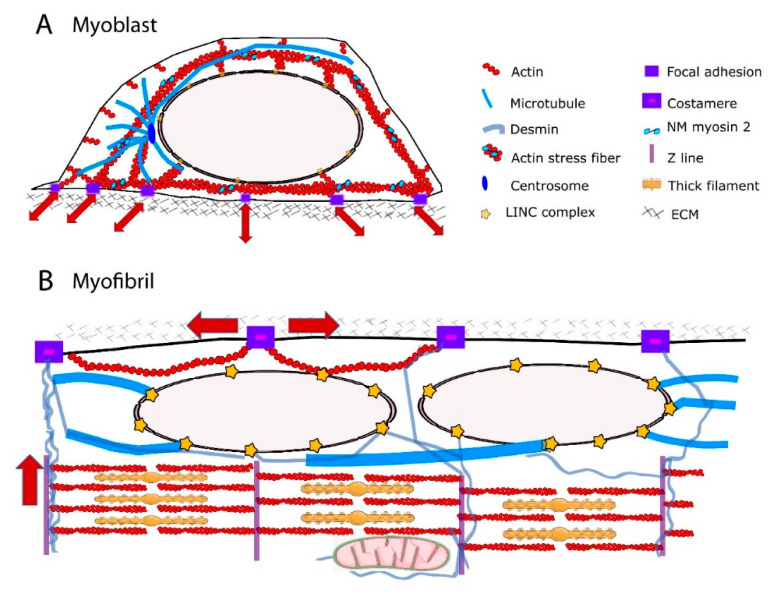
Schematic representation of cytoskeleton and force transmission in the myoblast and myofibril. (**A**) Radial distribution of the actin, microtubule and intermediate filament (IF) networks in myoblast favors the transmission of extra- and intra-cellular forces (red arrows) to the nucleus. Direct connections between focal adhesions and the actin cytoskeleton transmit the force along actin fibers towards the nucleus. Reciprocally, intracellular forces can be transmitted from the cell interior to the extracellular matrix (ECM). Perinuclear cytoskeleton is tethered to the nucleus via Linker of Nucleoskeleton and Cytoskeleton (LINC) complex. (**B**) Paraxial arrays of F-actin, microtubules and IFs in myofibrils. Main directions of force transmission from the contractile apparatus to the ECM are indicated (red arrows). In skeletal muscle, contractile force can be transmitted laterally between the z-disks of neighboring myofibrils to the ECM through specific cell–matrix adhesions called costameres.

**Figure 2 cells-10-00318-f002:**
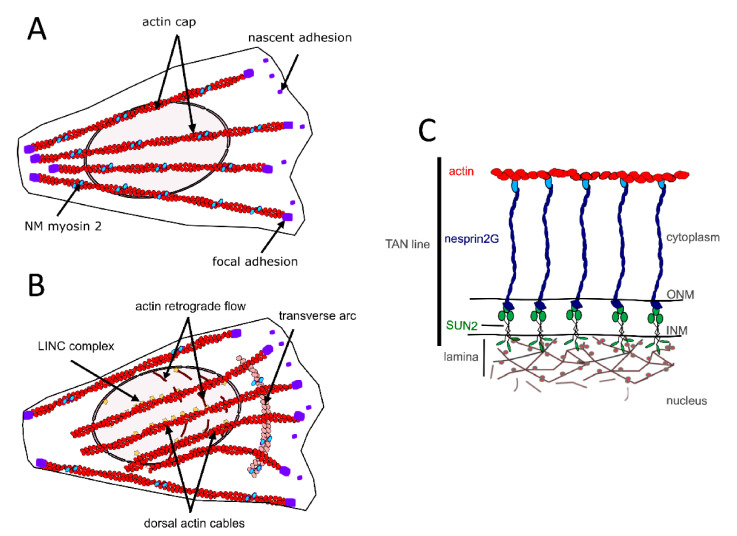
Components of the perinuclear actin network in muscle cell precursors (MCPs). (**A**) Actin cap formed by dorsal stress fibers. (**B**) transmembrane actin-associated nuclear (TAN) lines. (**C**) Illustration of the molecular composition of a TAN line.

**Figure 3 cells-10-00318-f003:**
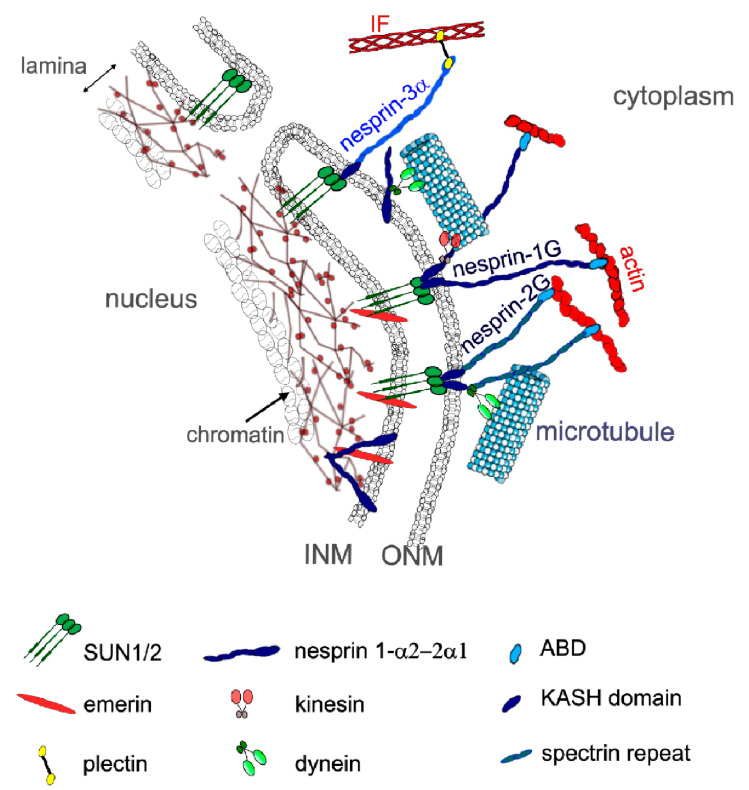
LINC complexes in skeletal muscle. LINC is a complex of proteins including SUN1/2 and nesprins that connect the cytoskeleton to the nucleoskeleton. Different nesprin isoforms are expressed during myogenesis: in MCPs, nesprin-1G and -2G can interact with actin and microtubules in the cytoplasm and with SUN1/2 proteins, emerin and lamins, on the inner nuclear membrane. Shorter nesprin-1α2 and nesprin-2α1 are expressed during myotube differentiation and can bind with microtubules in the cytoplasm via kinesin and other proteins such as A-kinase anchoring protein. Short nesprin-1α2 can also interact with intranuclear proteins such as lamins and emerin. INN: inner nuclear membrane; ONM: outer nuclear membrane.

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
