# Peer review of "Nuclear Mechanotransduction in Skeletal Muscle"

_cells, 2021, doi:10.3390/cells10020318_

Round 1

Reviewer 1 Report

In this interesting review Jabre and colleagues extensively reviewed general findings on nuclear mechanotransduction, with a special focus on skeletal muscle.

As a whole, the review is well written and references are updated.

I have just a couple of comments.

1) The review is mainly focused on the cytoskeleton, while only a very short paragraph covers the topic of chromatin-mediated mechanoresponse. Maybe authors could discuss in more detail the impact of nuclear mechanotransduction on the regulation gene expression.

2) With regard to Figure 2, I suggest to illustrate more clearly which part of the actin cytoskeleton is involved in the different force transmissions.

Author Response

We sincerely appreciate the reviewers’ comments and suggestions to help improve our manuscript. Each specific suggestion and comment have been addressed and incorporated into the revised version of the manuscript.

We thank Reviewer 1 for their time and effort on reviewing our paper and sincerely appreciate the positive comments and insightful suggestions.

Reviewer 2 Report

In their manuscript, Jabre et al. discuss and summarize recent developments in the field of nuclear mechanotransduction in skeletal muscle. Specifically, this review delves into our mechanistic understanding of how cytoplasmic mechanical forces are transmitted across the nuclear envelope in myoblasts and myotubes. In addition, the authors discuss how the cytoskeleton and nuclear interior are dynamically reorganized during myogenic differentiation and how these events impact nuclear mechanotransduction in healthy and diseased muscle. While this manuscript addresses an exciting and timely topic in the field, I have several major and minor issues that I feel need to be dealt with prior to my recommending that it be accepted for publication. These issues are outlined below.

Major Issues:

  • The authors reference many review articles at the expense of primary literature. While I realize that there may be limitations on the number of references that they can use in this manuscript, I would strongly suggest that the primary literature be referenced preferentially to the review articles.
  • Figure 1:
    1. The cell illustration presented in Panel A is drawn in a way that makes it very difficult to differentiate the intracellular structures apart from one another. The illustration may be too small. In addition, the authors might consider providing top-down views of the cells in panels A and B alongside the side-on views already present.
    2. It would be good for the microtubules and the centrosome to be drawn in different colors. The same could be said for the actin and dystrophin.
    3. What are the intranuclear structures shown in the nuclei in the illustrations of panels A and B?
    4. The ECM needs to be labeled either in the key or the illustrations shown in panels A and B.
    5. Why do the focal adhesions drawn in panel A not have integrins?
    6. Why does the nucleus in panel A lack LINC complexes?
    7. For clarity, the authors might consider just having one colored shape to represent the LINC complexes in the nuclei shown in panels A and B rather than the 3 differently colored shapes that already exist.
    8. Why do the authors draw all of the different components of the integrins and dystrophin-dystoglycan complexes but not the LINC complex nor the focal adhesions? Perhaps it would be best to only use differently colored shapes in an abstract manner rather than a mix as is currently the case?
    9. In the legend for panel B, the authors state, ­­"Perinuclear F-actin is tethered to the nucleus via LINC complex in myoblast”. However, panel B shows a myotube and neither a myofibril nor a myoblast. Moreover, LINC complexes tether nuclei to the perinuclear cytoskeleton in both myoblasts and myotubes, not just in myoblasts.
    10. The authors wrote “non-muscle myosin” in the figure legend. However, they should change it to “non-muscle myosin II”, as there are several non-muscle myosin proteins.
  • Figure 2:
    1. The organization of the perinuclear actin cytoskeleton drawn in Figure 2 is misleading, as TAN lines and the perinuclear actin cap are not known to be perpendicular to each other on the dorsal nuclear surface. The way that the authors have drawn the perinuclear actin cap is actually more reminiscent to the focal adhesion-capped stress fibers that exist on the ventral nuclear surface. This is because those stress fibers are perpendicular to the actin cables found on the dorsal nuclear surface and the direction of cell migration (see Luxton and Gomes et al., 2010). I would strongly recommend that the authors include a drawing of the cell perimeter in this illustration, as that geometric information would help the readers better understand the organization of perinuclear actin here. Moreover, TAN lines would be associated with a sub-population of the dorsal perinuclear actin cables that make up the perinuclear actin cap.
    2. It is insufficient to simply use “LINC” to label the LINC complexes that make up the TAN lines. TAN lines are linear arrays of nesprin-2G/SUN2-containing LINC complexes. SUN1 is not found in TAN lines (see Luxton and Gomes et al., 2010). In addition, the linear arrays of nesprin-2G/SUN2-containing LINC complexes typically extend across the entire dorsal nuclear surface, not as the isolated spotwelds drawn in this figure.
    3. The authors need to label non-muscle myosin II somewhere in the figure.
    4. The figure legend states, “Top view of the perinuclear actin network in MuSCs”. Yet, the authors never explained what “MuSCs” are. I assume that these are myoblasts?
    5. I would recommend that the authors clean up their nuclear envelope illustration here, as it seems to have been duplicated in the upper right hand corner.
  • Figure 3:
    1. The way kinesin and dynein are drawn makes it seem like they are the same when in fact they are not. I would recommend drawing them as such.
    2. There is no evidence to suggest that nesprin-1 or nesprin-2 associate with kinesin or dynein via their N-termini. Both motors seem to associate with these nesprins near the C-terminal portion of their cytoplasmic domains (i.e. near the transmembrane domain).
    3. The nesprin-1 and nesprin-2 proteins that the authors have drawn bound to the actin cytoskeleton should be labeled nesprin-1G and nesprin-2G, respectively. I would also recommend making their N-terminal actin-binding domains a distinct color.
    4. The association of plectin with nesprin-3 occurs via the 1st spectrin-like repeat, which is how the authors have drawn the association for the nesprin-3 protein that they have labeled. However, I am unaware of any association between the middle spectrin-like repeats of nesprin-3 and plectin. Nor am I aware of an interaction between nesprin-3 and actin. Thus, the authors need to do the following:
      1. Rename “nesprin-3” as “nesprin-3α”.
      2. Remove the nesprin-3 bound to plectin and actin at the bottom of the illustration.
    5. I am unaware of any evidence to support the way that the authors have drawn the nucleoplasm-facing nesprin-1-α2/-2α1 in complex with a homo-trimer of SUN1/2. While it is known that nesprins can be found on the inner nuclear membrane, this is most likely due to them having a transmembrane domain and not due to their association with a SUN1/2 homo-trimer.
    6. The way that the authors have drawn the nuclear lamins is inconsistent with their mesh-like appearance as revealed by recent super-resolution light microscopy and cryo-EM tomography (see recent work from the Goldman and Medalia labs).
    7. Since the KASH peptides of most nesprins are ~20-30 amino acids in length, the SUN1/2 homo-trimers most likely need to extend across the entire perinuclear space to the outer nuclear membrane to be able to form a LINC complex. I would recommend that the authors increase the size of the luminal domain of their SUN1/SUN2 trimers such that they extend out to the outer nuclear envelope and decrease the size of their KASH peptides accordingly.
  • I find it strange that the authors neglect to discuss the potential roles of nuclear mechanotransduction during muscle injury repair. Myonuclei are known to move inwards following injury in a manner that is similar to what occurs in developing muscle (see Folker and Baylies, 2013 Front Physiol).
  • Lines 123-125: The authors state that myoblasts are “non-contractile”; however, this is not true. Myoblast contain actomyosin and are thus contractile. Therefore, there needs to be a better way for the authors to differentiate between myoblast and myofibers. Alternatively, the authors could simply remove “non-contractile” and “contractile” as descriptions for myoblasts and myotubes, respectively.
  • Lines 110-111: The statement that “parallel actomyosin cables present in the perinuclear region of myotubes are not physically tethered to the nucleus” made by the authors does not seem to have experimental evidence to back it up. This statement was made based on the Wang et al. (2015) J Cell Biol paper, which describes how the nesprin protein MSP300 works together with spectraplakin and EB1 to regulate the elastic properties of muscle nuclei in Drosophila. Yet, the authors of this manuscript neglect to mention that this work was performed in flies. Since nuclear-cytoskeletal coupling in fly and mammalian muscles may be regulated through different mechanisms, I would strongly encourage the authors to explicitly make this distinction.
  • Lines 213-214: The authors state that plectin can interact with nesprin-1 and -2 in addition to nesprin-3α. However, I am unaware of these interactions and was unable to find evidence to support them in the references provided by the authors.
  • Lines 220-221: The authors refer their readers to reference 145 as support for the statement, “In human(s), complete depletion of the giant nesprin-1 is responsible for severe congenital muscular dystrophy”. While this paper reports the results of experiments performed in human myoblasts, it by no means shows that nesprin-2-depletion causes severe congenital muscular dystrophy.
  • Lines 245-246: The authors state, “B-type lamins do not appear to play a major role in nuclear stiffness”. However, recently published work (Gill et al., 2019 Front Cell Devel Biol) shows that mouse embryonic fibroblasts lacking lamin B1 or B2 are more deformable than controls. Thus, their statement is not completely correct and their text needs to be adjusted accordingly.

Minor Issues:

  • Please define all abbreviations the first time that they are used in the manuscript.
    1. MuSCs, YAP, TAZ, ATR, cPLA2, SUN, nesprin, AKAP-450, WFS1
    2. If an abbreviation is used only once, there is no need to use an abbreviation (e.g. “L-CMD”).
  • There are numerous examples of subject-verb agreement errors throughout the manuscript that need to be addressed.
    1. Line 38: “MCPs” need to be changed to “MCP”.
    2. Line 138: “confers” need to be changed to “confer”.
    3. Line 201: “are” and “become” need to be changed to “is” and “becomes”, respectively.
    4. Line 212: “IF” needs to be changed to “IFs”.
    5. Line 220: “human” needs to be changed to “humans”.
    6. Line 237: “provide” needs to be changed to “provides”.
    7. Line 247: “dictate” needs to be changed to “dictates”.
    8. Line 258: “lamin” needs to be changed to “lamins”.
    9. Line 329: “response” needs to be changed to “responses”.
  • There are several inappropriately used “the”’s used throughout the manuscript.
    1. Line 14: Delete the “the”.
    2. Line 126: Delete the “the” and capitalize “microtubules”.
    3. Line 142: Delete the “the” and capitalize “Cytoplasmic”.
    4. Line 173: Delete the “The” before “LINC”.
    5. Line 209: Delete the “The” before “nesprin-3” and capitalize “nesprin-3”.
    6. Line 261: Delete the “the” before “myogenic”.
    7. Line 326: Insert a “the” before “terminal”.
  • Mammalian gene names are capitalized and written in italics.
    1. Lines 182 and 185: SYNE1 and SYNE2.
  • Line 61: It is unclear what “the mechanical environment” refers to in this line. Is it the cellular or extracellular mechanical environment?
  • Line 100: The “dorsal side of the nucleus” is unclear in this line. I assume that you are referring to cells grown in 2D culture? If so, “dorsal” need to be defined in relation to the substrate/coverglass that the cell is grown on.
  • Line 104: What do the authors mean by “at the central part” in this line? Are they referring to the “central part” of a stress fiber?
  • Line 112: The authors need to explain what the “spectraplakin-EB1-microtubule complex” is and does.
  • Line 127: Here, the authors discuss the stiffness of microtubules in relation to actin while ignoring intermediate filaments. While I realize that the authors devote the next section of their manuscript to cytoplasmic intermediate filaments, the line in question makes it seem like the actin and microtubule cytoskeletons are the only components of the cytoskeleton. This is clearly not true. The authors should discuss the bending stiffness of the cytoplasmic intermediate filaments, which is lower than either actin or microtubule bending stiffness. This is due to the short persistence length of intermediate filaments (1-3 μm).
  • Line 130: The authors need to describe what “MT nucleating material” means.
  • Line 135: It is unclear what “reduced tyrosinated/detyrosinated” means here. The authors should simply state “increased detyrosination”.
  • Line 144: What do the authors mean by “unique mechanical stiffness property” exactly?
  • Line 145: How large are the “large deformations” referenced here?
  • Line 156: Delete the first “nuclear”, as it is redundant to write “nuclear outer nuclear membrane”.
  • Line 147: Reference 82 is inappropriately used here, as it does not discuss the interaction of cytoplasmic intermediate filaments to other cytoskeletal filaments or membrane complexes in cells.
  • Lines 156-157: Reference 91 did not show that vimentin and desmin linked to the outer nuclear membrane “via a class of large proteins related to spectrin and plakins”. Thus, this statement is incorrect.
  • Line 175: The authors should insert “C-terminal” in between “their” and “the”.
  • Line 176: The authors need to replace “ANC” with “ANC-1” and they need to delete “at the C-terminals”. In addition, I would suggest that they replace “This domain binds to perinuclear SUN proteins 1 and 2” with “This domain directly interacts with the luminal domain of the inner nuclear membrane proteins SUN1 or SUN2 within the perinuclear space of the nuclear envelope”.
  • Line 178: I would recommend that the authors replace “on the nucleoplasmic side” with “within the nucleoplasm”, as “nucleoplasmic side” is unclear.
  • Line 179: I would recommend that the authors replace “cytoplasm to the inner nucleus” with “cytoskeleton to the nucleoskeleton”, as “inner nucleus” is unclear.
  • Lines 180-181: The authors need references for this statement.
  • Lines 183 and 220-221: The authors should attach “G” to “nesprin-1” and “nesprin-2”, as this is the convention in the field for referring to the giant nesprin isoforms.
  • Line 184: The authors should replace “N-terminials” with “N-termini”, as this is convention in the field.
  • Line 185: The phrase “via their cytoplasmic stretch” communicates nothing about what we know regarding the interaction of nesprin-1 and -2 with the microtubule motor proteins dynein or kinesin. For example, kinesin-1 interacts with nesprin-1 and -2 via their LEWD motifs, which is present within their adaptive domains. The authors need to be more precise in their writing.
  • Lines 186-187: While the authors are correct that SYNE1 and SYNE2 have multiple internal promoters that give rise to shorter nesprin isoforms that lack the N-terminal CH domains, alternative splicing also generates short isoforms that lack the C-terminal KASH domain as well as those that lack both the KASH domain and CH domains. Thus, this text needs to be adjusted accordingly.
  • Line 200: The authors should insert “ability to form” in between “its” and “interactions”.
  • Line 206: Delete “protein” from this line, as its inclusion does not make sense.
  • Line 211: The phrase “an actin-binding domain” used here needs to be changed to “tandem actin-binding calponin homology domains” to be more specific.
  • Lines 211-212: The authors need to explain how nesprin-3α and -3β differ from each other. Specifically, nesprin-3β lacks the 1st spectrin-like repeat of nesprin-3α, which interacts with plectin.
  • Lines 213-214: The authors fail to acknowledge the fact that plectin can crosslink the actin, intermediate filament, and microtubule cytoskeletons, not just “actin filaments to desmin”. This information is important to consider when thinking about mechanotransduction.
  • Line 219: The authors need to insert “Emery Dreifuss” before “muscle” and they need to change “muscle” to “muscular”.
  • Line 218: The authors need to include Luxton and Gomes et al. (2010) Science as a reference for the statement that LINC complex inhibition “altered centrosome positioning”. However, they should be aware of the fact that LINC complex inhibition does not inhibit centrosome positioning in migrating fibroblasts or myoblasts. Rather, inhibiting the LINC complex blocks rearward nuclear movement by actin retrograde flow, preventing the orientation of the centrosome towards the direction of migration in fibroblasts and myoblasts. That being said, there are examples of the depletion of LINC complex components (e.g. nesprin-2) preventing the migration of centrosomes to the plasma membranes during ciliogenesis (Dawe et al., 2009 J Cell Sci).
  • Line 233: The authors should provide their readers with an example of said “LINC-independent mechanisms”.
  • Line 249: The authors should insert “nucleoplasmic domain of” between “the” and “inner”.
  • Line 264: Insert “levels of” between “the” and “repressive”.
  • Line 273: It is unclear to me what “NE” means here.
  • Line 285: The “/” should be changed to “, “ and a “,” should be added after “factors”.
  • Line 291: Change “to” to “that”.
  • Line 294: Insert a “the” before “altered”.
  • Line 328: Attach a “-“ to the end of “stress”.

Author Response

We sincerely appreciate the reviewers’ comments and suggestions to help improve our manuscript. Each specific suggestion and comment have been addressed below and incorporated into the revised version of the manuscript.

Reviewer 2

We sincerely thank Reviewer 2 for careful reading and effort on reviewing our paper and sincerely appreciate the positive comments and insightful suggestions. The following are our point-by-point responses.

Comments and Suggestions for Authors

In their manuscript, Jabre et al. discuss and summarize recent developments in the field of nuclear mechanotransduction in skeletal muscle. Specifically, this review delves into our mechanistic understanding of how cytoplasmic mechanical forces are transmitted across the nuclear envelope in myoblasts and myotubes. In addition, the authors discuss how the cytoskeleton and nuclear interior are dynamically reorganized during myogenic differentiation and how these events impact nuclear mechanotransduction in healthy and diseased muscle. While this manuscript addresses an exciting and timely topic in the field, I have several major and minor issues that I feel need to be dealt with prior to my recommending that it be accepted for publication. These issues are outlined below.

Major Issues:

  1. The authors reference many review articles at the expense of primary literature. While I realize that there may be limitations on the number of references that they can use in this manuscript, I would strongly suggest that the primary literature be referenced preferentially to the review articles.

This comment has been taken into account whenever possible.

  1. Figure 1:
    1. The cell illustration presented in Panel A is drawn in a way that makes it very difficult to differentiate the intracellular structures apart from one another. The illustration may be too small. In addition, the authors might consider providing top-down views of the cells in panels A and B alongside the side-on views already present.

All comments have been taken into account: the illustration is bigger, side-on views have been added in the Revised Figure 2.

    1. It would be good for the microtubules and the centrosome to be drawn in different colors. The same could be said for the actin and dystrophin.

This has been done

    1. What are the intranuclear structures shown in the nuclei in the illustrations of panels A and B?

Intranuclear structures represent chromatin

    1. The ECM needs to be labeled either in the key or the illustrations shown in panels A and B.

This has been done

    1. Why do the focal adhesions drawn in panel A not have integrins?

Focal adhesions have been re-drawn

    1. Why does the nucleus in panel A lack LINC complexes?

LINC complexes have been added

    1. For clarity, the authors might consider just having one colored shape to represent the LINC complexes in the nuclei shown in panels A and B rather than the 3 differently colored shapes that already exist.

This has been done

    1. Why do the authors draw all of the different components of the integrins and dystrophin-dystoglycan complexes but not the LINC complex nor the focal adhesions? Perhaps it would be best to only use differently colored shapes in an abstract manner rather than a mix as is currently the case?

The Figure has been modified, as recommended.

    1. In the legend for panel B, the authors state, ­­"Perinuclear F-actin is tethered to the nucleus via LINC complex in myoblast”. However, panel B shows a myotube and neither a myofibril nor a myoblast. Moreover, LINC complexes tether nuclei to the perinuclear cytoskeleton in both myoblasts and myotubes, not just in myoblasts.

The legend has been modified, as recommended.

    1. The authors wrote “non-muscle myosin” in the figure legend. However, they should change it to “non-muscle myosin II”, as there are several non-muscle myosin proteins.—

This has been modified

  1. Figure 2:
    1. The organization of the perinuclear actin cytoskeleton drawn in Figure 2 is misleading, as TAN lines and the perinuclear actin cap are not known to be perpendicular to each other on the dorsal nuclear surface. The way that the authors have drawn the perinuclear actin cap is actually more reminiscent to the focal adhesion-capped stress fibers that exist on the ventral nuclear surface. This is because those stress fibers are perpendicular to the actin cables found on the dorsal nuclear surface and the direction of cell migration (see Luxton and Gomes et al., 2010). I would strongly recommend that the authors include a drawing of the cell perimeter in this illustration, as that geometric information would help the readers better understand the organization of perinuclear actin here. Moreover, TAN lines would be associated with a sub-population of the dorsal perinuclear actin cables that make up the perinuclear actin cap.

Figure has been modified as required. Additional panels have been added to illustrate the different actin cytoskeleton and TAN lines. We hope this clarify.

    1. It is insufficient to simply use “LINC” to label the LINC complexes that make up the TAN lines. TAN lines are linear arrays of nesprin-2G/SUN2-containing LINC complexes. SUN1 is not found in TAN lines (see Luxton and Gomes et al., 2010). In addition, the linear arrays of nesprin-2G/SUN2-containing LINC complexes typically extend across the entire dorsal nuclear surface, not as the isolated spotwelds drawn in this figure.

Figure has been modified as recommend and additional panels have been added.

    1. The authors need to label non-muscle myosin II somewhere in the figure.

This has been done

    1. The figure legend states, “Top view of the perinuclear actin network in MuSCs”. Yet, the authors never explained what “MuSCs” are. I assume that these are myoblasts?

Legend has been modified as recommended.

    1. I would recommend that the authors clean up their nuclear envelope illustration here, as it seems to have been duplicated in the upper right hand corner.

This has been done.

  1. Figure 3:
    1. The way kinesin and dynein are drawn makes it seem like they are the same when in fact they are not. I would recommend drawing them as such.

This has been done

    1. There is no evidence to suggest that nesprin-1 or nesprin-2 associate with kinesin or dynein via their N-termini. Both motors seem to associate with these nesprins near the C-terminal portion of their cytoplasmic domains (i.e. near the transmembrane domain).

Figure has been modified as recommended.

    1. The nesprin-1 and nesprin-2 proteins that the authors have drawn bound to the actin cytoskeleton should be labeled nesprin-1G and nesprin-2G, respectively. I would also recommend making their N-terminal actin-binding domains a distinct color.

Figure has been modified as recommended.

    1. The association of plectin with nesprin-3 occurs via the 1st spectrin-like repeat, which is how the authors have drawn the association for the nesprin-3 protein that they have labeled. However, I am unaware of any association between the middle spectrin-like repeats of nesprin-3 and plectin. Nor am I aware of an interaction between nesprin-3 and actin. Thus, the authors need to do the following:
      1. Rename “nesprin-3” as “nesprin-3α”.
      2. Remove the nesprin-3 bound to plectin and actin at the bottom of the illustration.

Figure has been modified as recommended.

    1. I am unaware of any evidence to support the way that the authors have drawn the nucleoplasm-facing nesprin-1-α2/-2α1 in complex with a homo-trimer of SUN1/2. While it is known that nesprins can be found on the inner nuclear membrane, this is most likely due to them having a transmembrane domain and not due to their association with a SUN1/2 homo-trimer.

Figure has been modified as recommended.

    1. The way that the authors have drawn the nuclear lamins is inconsistent with their mesh-like appearance as revealed by recent super-resolution light microscopy and cryo-EM tomography (see recent work from the Goldman and Medalia labs).

Figure has been revised as recommended.

    1. Since the KASH peptides of most nesprins are ~20-30 amino acids in length, the SUN1/2 homo-trimers most likely need to extend across the entire perinuclear space to the outer nuclear membrane to be able to form a LINC complex. I would recommend that the authors increase the size of the luminal domain of their SUN1/SUN2 trimers such that they extend out to the outer nuclear envelope and decrease the size of their KASH peptides accordingly.

Figure has been revised as recommended.

  1. I find it strange that the authors neglect to discuss the potential roles of nuclear mechanotransduction during muscle injury repair. Myonuclei are known to move inwards following injury in a manner that is similar to what occurs in developing muscle (see Folker and Baylies, 2013 Front Physiol).

This has been done.

  1. Lines 123-125: The authors state that myoblasts are “non-contractile”; however, this is not true. Myoblast contain actomyosin and are thus contractile. Therefore, there needs to be a better way for the authors to differentiate between myoblast and myofibers. Alternatively, the authors could simply remove “non-contractile” and “contractile” as descriptions for myoblasts and myotubes, respectively.

This has been modified.

  1. Lines 110-111: The statement that “parallel actomyosin cables present in the perinuclear region of myotubes are not physically tethered to the nucleus” made by the authors does not seem to have experimental evidence to back it up. This statement was made based on the Wang et al. (2015) J Cell Biol paper, which describes how the nesprin protein MSP300 works together with spectraplakin and EB1 to regulate the elastic properties of muscle nuclei in Drosophila. Yet, the authors of this manuscript neglect to mention that this work was performed in flies. Since nuclear-cytoskeletal coupling in fly and mammalian muscles may be regulated through different mechanisms, I would strongly encourage the authors to explicitly make this distinction.

The paragraph has been revised as recommended.

  1. Lines 213-214: The authors state that plectin can interact with nesprin-1 and -2 in addition to nesprin-3α. However, I am unaware of these interactions and was unable to find evidence to support them in the references provided by the authors.

This has been corrected

  1. Lines 220-221: The authors refer their readers to reference 145 as support for the statement, “In human(s), complete depletion of the giant nesprin-1 is responsible for severe congenital muscular dystrophy”. While this paper reports the results of experiments performed in human myoblasts, it by no means shows that nesprin-2-depletion causes severe congenital muscular dystrophy.

We agree with this comment and we did not claim that nesprin-2 depletion causes congenital muscular dystrophy. We wrote “It has been shown that mutations in nesprins-1 and -2 cause muscle dystrophy [73,141–145] and dilated cardiomyopathy [144]. In human, complete depletion of the giant nesprin-1 is responsible for severe congenital muscular dystrophy [145]. »

  1. Lines 245-246: The authors state, “B-type lamins do not appear to play a major role in nuclear stiffness”. However, recently published work (Gill et al., 2019 Front Cell Devel Biol) shows that mouse embryonic fibroblasts lacking lamin B1 or B2 are more deformable than controls. Thus, their statement is not completely correct and their text needs to be adjusted accordingly.

The paragraph has been rewritten

Minor Issues:

  • Please define all abbreviations the first time that they are used in the manuscript.
    1. MuSCs, YAP, TAZ, ATR, cPLA2, SUN, nesprin, AKAP-450, WFS1
    2. If an abbreviation is used only once, there is no need to use an abbreviation (e.g. “L-CMD”).

This has been done

  • There are numerous examples of subject-verb agreement errors throughout the manuscript that need to be addressed.

All typos have been corrected, as recommended, including:

    1. Line 38: “MCPs” need to be changed to “MCP”.
    2. Line 138: “confers” need to be changed to “confer”.
    3. Line 201: “are” and “become” need to be changed to “is” and “becomes”, respectively.
    4. Line 212: “IF” needs to be changed to “IFs”.
    5. Line 220: “human” needs to be changed to “humans”.
    6. Line 237: “provide” needs to be changed to “provides”.
    7. Line 247: “dictate” needs to be changed to “dictates”. A-type lamins… dictate the nuclear…
    8. Line 258: “lamin” needs to be changed to “lamins”.
    9. Line 329: “response” needs to be changed to “responses”.

  • There are several inappropriately used “the”’s used throughout the manuscript.
    1. Line 14: Delete the “the”.
    2. Line 126: Delete the “the” and capitalize “microtubules”.
    3. Line 142: Delete the “the” and capitalize “Cytoplasmic”.
    4. Line 173: Delete the “The” before “LINC”.
    5. Line 209: Delete the “The” before “nesprin-3” and capitalize “nesprin-3”.
    6. Line 261: Delete the “the” before “myogenic”.
    7. Line 326: Insert a “the” before “terminal”.

The inappropriated use of “the” have been corrected, as recommended.

Mammalian gene names have to be checked.

Mammalian gene names are capitalized and written in italics.

Lines 182 and 185: SYNE1 and SYNE2.

  • Line 61: It is unclear what “the mechanical environment” refers to in this line. Is it the cellular or extracellular mechanical environment?

This has been clarified

  • Line 100: The “dorsal side of the nucleus” is unclear in this line. I assume that you are referring to cells grown in 2D culture? If so, “dorsal” need to be defined in relation to the substrate/coverglass that the cell is grown on.

This has been clarified

  • Line 104: What do the authors mean by “at the central part” in this line? Are they referring to the “central part” of a stress fiber?

The sentence has been modified.

  • Line 112: The authors need to explain what the “spectraplakin-EB1-microtubule complex” is and does.

The paragraph has been modified.

  • Line 127: Here, the authors discuss the stiffness of microtubules in relation to actin while ignoring intermediate filaments. While I realize that the authors devote the next section of their manuscript to cytoplasmic intermediate filaments, the line in question makes it seem like the actin and microtubule cytoskeletons are the only components of the cytoskeleton. This is clearly not true. The authors should discuss the bending stiffness of the cytoplasmic intermediate filaments, which is lower than either actin or microtubule bending stiffness. This is due to the short persistence length of intermediate filaments (1-3 μm).

The bending stiffness of IF have been added, as recommended.

  • Line 130: The authors need to describe what “MT nucleating material” means.

This has been done

  • Line 135: It is unclear what “reduced tyrosinated/detyrosinated” means here. The authors should simply state “increased detyrosination”.

This has been done

  • Line 144: What do the authors mean by “unique mechanical stiffness property” exactly?

The unique mechanical stiffness property has been clarified

  • Line 145: How large are the “large deformations” referenced here?

This has been clarified

  • Line 156: Delete the first “nuclear”, as it is redundant to write “nuclear outer nuclear membrane”.

This has been done

  • Line 147: Reference 82 is inappropriately used here, as it does not discuss the interaction of cytoplasmic intermediate filaments to other cytoskeletal filaments or membrane complexes in cells.

This has been done

  • Lines 156-157: Reference 91 did not show that via a class of large proteins related to spectrin and plakins “via a class of large proteins related to spectrin and plakins”. Thus, this statement is incorrect.

The paragraph has been modified

  • Line 175: The authors should insert “C-terminal” in between “their” and “the”.

This has been done

  • Line 176: The authors need to replace “ANC” with “ANC-1” and they need to delete “at the C-terminals”. In addition, I would suggest that they replace “This domain binds to perinuclear SUN proteins 1 and 2” with “This domain directly interacts with the luminal domain of the inner nuclear membrane proteins SUN1 or SUN2 within the perinuclear space of the nuclear envelope”.

This has been done

  • Line 178: I would recommend that the authors replace “on the nucleoplasmic side” with “within the nucleoplasm”, as “nucleoplasmic side” is unclear.

This has been done

  • Line 179: I would recommend that the authors replace “cytoplasm to the inner nucleus” with “cytoskeleton to the nucleoskeleton”, as “inner nucleus” is unclear.

This has been done

  • Lines 180-181: The authors need references for this statement.

This has been done

  • Lines 183 and 220-221: The authors should attach “G” to “nesprin-1” and “nesprin-2”, as this is the convention in the field for referring to the giant nesprin isoforms.

This has been done

  • Line 184: The authors should replace “N-terminials” with “N-termini”, as this is convention in the field.

This has been done

  • Line 185: The phrase “via their cytoplasmic stretch” communicates nothing about what we know regarding the interaction of nesprin-1 and -2 with the microtubule motor proteins dynein or kinesin. For example, kinesin-1 interacts with nesprin-1 and -2 via their LEWD motifs, which is present within their adaptive domains. The authors need to be more precise in their writing.

Details have been provided

  • Lines 186-187: While the authors are correct that SYNE1 and SYNE2 have multiple internal promoters that give rise to shorter nesprin isoforms that lack the N-terminal CH domains, alternative splicing also generates short isoforms that lack the C-terminal KASH domain as well as those that lack both the KASH domain and CH domains. Thus, this text needs to be adjusted accordingly.

This has been done

  • Line 200: The authors should insert “ability to form” in between “its” and “interactions”.

This has been done.

  • Line 206: Delete “protein” from this line, as its inclusion does not make sense.

This has been done.

  • Line 211: The phrase “an actin-binding domain” used here needs to be changed to “tandem actin-binding calponin homology domains” to be more specific.

This has been done.

  • Lines 211-212: The authors need to explain how nesprin-3α and -3β differ from each other. Specifically, nesprin-3β lacks the 1st spectrin-like repeat of nesprin-3α, which interacts with plectin.

This has been done.

  • Lines 213-214: The authors fail to acknowledge the fact that plectin can crosslink the actin, intermediate filament, and microtubule cytoskeletons, not just “actin filaments to desmin”. This information is important to consider when thinking about mechanotransduction.

This has been done.

  • Line 219: The authors need to insert “Emery Dreifuss” before “muscle” and they need to change “muscle” to “muscular”.

This has been done.

  • Line 218: The authors need to include Luxton and Gomes et al. (2010) Science as a reference for the statement that LINC complex inhibition “altered centrosome positioning”. However, they should be aware of the fact that LINC complex inhibition does not inhibit centrosome positioning in migrating fibroblasts or myoblasts. Rather, inhibiting the LINC complex blocks rearward nuclear movement by actin retrograde flow, preventing the orientation of the centrosome towards the direction of migration in fibroblasts and myoblasts. That being said, there are examples of the depletion of LINC complex components (e.g. nesprin-2) preventing the migration of centrosomes to the plasma membranes during ciliogenesis (Dawe et al., 2009 J Cell Sci).

This has been done.

  • Line 233: The authors should provide their readers with an example of said “LINC-independent mechanisms”.

This has been done.

  • Line 249: The authors should insert “nucleoplasmic domain of” between “the” and “inner”.

This has been done.

  • Line 264: Insert “levels of” between “the” and “repressive”.

This has been done.

  • Line 273: It is unclear to me what “NE” means here.

This has been modified

  • Line 285: The “/” should be changed to “, “ and a “,” should be added after “factors”.

This has been done

  • Line 291: Change “to” to “that”.

This has been done

  • Line 294: Insert a “the” before “altered”.

This has been done

  • Line 328: Attach a “-“ to the end of “stress”.

This has been done

Round 2

Reviewer 2 Report

I thank the authors for addressing my previously identified criticisms regarding their original manuscript. Overall, they have largely assuaged the majority of my concerns. However, I do suggest the following changes be made prior to publication of this review article.

  • Figure 1:
    1. The authors need to define for their readers the intranuclear structures drawn in panels A and B. I assume that the light red shapes found at the nuclear periphery represent heterochromatin, but I do not know what the dark purple shape is in the middle of the nucleus. A nucleolus?
    2. The authors should delete the rogue LINC complex drawn to the right of the right myonucleus presented in panel B.
    3. The authors should also draw the LINC complexes as being evenly distributed throughout the nuclear envelopes present in this figure. The myonuclei in panel B seem to have a polarized LINC complex distribution.

  • Figure 2:
    1. In panel B, the TAN lines are drawn as short actin tetramers on the dorsal nuclear surface. However, TAN lines are linear arrays of LINC complexes that are associated with perinuclear actin cables that are found on the dorsal nuclear surface that stretch across the width of the cell. Thus, they should re-draw the perinuclear actin cables accordingly and show then being associated with linear arrays of the LINC complex-representing star shapes.
    2. The nuclear lamins need to be labeled in panel C.

  • Figure 3:
    1. The authors need to label the chromatin in this illustration.
    2. The way that the SUN protein homo-trimers are drawn makes it seem like their SUN domains extend the length of the perinuclear space. The authors should make the trimer smaller and include part of the SUN protein luminal domain between the transmembrane domain and the SUN domains (e.g. a coiled-coil).
    3. It might be better to redraw the microtubules so that only one runs parallel to the nuclear envelope, rather than having two microtubules that are perpendicular to the nucleus.
    4. There is no evidence to support the existence of nesprin proteins that span the perinuclear space. The authors should redraw their “nesprin 1-alpha2-2alpha1” proteins accordingly. One should be protruding out from the outer nuclear membrane into the cytoplasm, while the other should protrude from the inner nuclear membrane into the nucleoplasm.

  • Line 133: “N-wasp” should be re-written as “N-WASP”.
  • Regarding my previous comment: “Lines 220-221: The authors refer their readers to reference 145 as support for the statement, “In human(s), complete depletion of the giant nesprin-1 is responsible for severe congenital muscular dystrophy”. While this paper reports the results of experiments performed in human myoblasts, it by no means shows that nesprin-2-depletion causes severe congenital muscular dystrophy.”
    1. My apologies for confusing nesprin-1 with nesprin-2 in my previous comment. However, reference 150 (Schwartz, Fischer et al., 2017) does not show that “complete depletion of the nesprin-1G is responsible for severe congenital muscular dystrophy”.

Author Response

We sincerely thank Reviewer 2 for time and effort on reviewing our paper and sincerely appreciate the comments and insightful suggestions. The following are our point-by-point responses. All comments have been taken into account in the revised version.

  • Figure 1:
    1. The authors need to define for their readers the intranuclear structures drawn in panels A and B. I assume that the light red shapes found at the nuclear periphery represent heterochromatin, but I do not know what the dark purple shape is in the middle of the nucleus. A nucleolus?

We agree that drawn intranuclear structures (i.e., chromatin and nucleolus) could be confusing in panels A and B. They have been suppressed in the revised Figure 1.

    1. The authors should delete the rogue LINC complex drawn to the right of the right myonucleus presented in panel B.

This has been done.

    1. The authors should also draw the LINC complexes as being evenly distributed throughout the nuclear envelopes present in this figure. The myonuclei in panel B seem to have a polarized LINC complex distribution.

 This has been done.

  • Figure 2:
    1. In panel B, the TAN lines are drawn as short actin tetramers on the dorsal nuclear surface. However, TAN lines are linear arrays of LINC complexes that are associated with perinuclear actin cables that are found on the dorsal nuclear surface that stretch across the width of the cell. Thus, they should re-draw the perinuclear actin cables accordingly and show then being associated with linear arrays of the LINC complex-representing star shapes.

 This has been done.

    1. The nuclear lamins need to be labeled in panel C.

  This has been done.

  • Figure 3:
    1. The authors need to label the chromatin in this illustration.

  This has been done.

    1. The way that the SUN protein homo-trimers are drawn makes it seem like their SUN domains extend the length of the perinuclear space. The authors should make the trimer smaller and include part of the SUN protein luminal domain between the transmembrane domain and the SUN domains (e.g. a coiled-coil).

  This has been done.

    1. It might be better to redraw the microtubules so that only one runs parallel to the nuclear envelope, rather than having two microtubules that are perpendicular to the nucleus.

  This has been done.

    1. There is no evidence to support the existence of nesprin proteins that span the perinuclear space. The authors should redraw their “nesprin 1-alpha2-2alpha1” proteins accordingly. One should be protruding out from the outer nuclear membrane into the cytoplasm, while the other should protrude from the inner nuclear membrane into the nucleoplasm.

   This has been done.

  • Line 133: “N-wasp” should be re-written as “N-WASP”.

This has been corrected.

  • Regarding my previous comment: “Lines 220-221: The authors refer their readers to reference 145 as support for the statement, “In human(s), complete depletion of the giant nesprin-1 is responsible for severe congenital muscular dystrophy”. While this paper reports the results of experiments performed in human myoblasts, it by no means shows that nesprin-2-depletion causes severe congenital muscular dystrophy.”

My apologies for confusing nesprin-1 with nesprin-2 in my previous comment. However, reference 150 (Schwartz, Fischer et al., 2017) does not show that “complete depletion of the nesprin-1G is responsible for severe congenital muscular dystrophy”.

We agree with this comment. The fact that complete deletion of the giant nesprin-1 is responsible for severe congenital muscular dystrophy has been reported in 2007 as an abstract (Voit T, et al. Congenital muscular dystrophy with adducted thumbs, mental retardation, cerebellar hypoplasia and cataracts is caused by mutation of Enaptin (Nesprin-1): The third nuclear envelopathy with muscular dystrophy. Neuromuscul Disord. 2007;17:833–834) but not follows by full article. Therefore, the claim has been deleted.